# The Effects of Hypoxia and Inflammation on Synaptic Signaling in the CNS

**DOI:** 10.3390/brainsci6010006

**Published:** 2016-02-17

**Authors:** Gatambwa Mukandala, Ronan Tynan, Sinead Lanigan, John J. O’Connor

**Affiliations:** UCD School of Biomolecular and Biomedical Science, UCD Conway Institute of Biomolecular and Biomedical Research, Belfield, Dublin 4, Ireland; Gatambwa.mukandala@ucdconnect.ie (G.M.); ronan.tynan@ucdconnect.ie (R.T.); sinead.lanigan@ucdconnect.ie (S.L.)

**Keywords:** hypoxia, TNF-α, adenosine, HIF-1α, hippocampus, long-term potentiation, prolyl hydroxylase inhibitor

## Abstract

Normal brain function is highly dependent on oxygen and nutrient supply and when the demand for oxygen exceeds its supply, hypoxia is induced. Acute episodes of hypoxia may cause a depression in synaptic activity in many brain regions, whilst prolonged exposure to hypoxia leads to neuronal cell loss and death. Acute inadequate oxygen supply may cause anaerobic metabolism and increased respiration in an attempt to increase oxygen intake whilst chronic hypoxia may give rise to angiogenesis and erythropoiesis in order to promote oxygen delivery to peripheral tissues. The effects of hypoxia on neuronal tissue are exacerbated by the release of many inflammatory agents from glia and neuronal cells. Cytokines, such as TNF-α, and IL-1β are known to be released during the early stages of hypoxia, causing either local or systemic inflammation, which can result in cell death. Another growing body of evidence suggests that inflammation can result in neuroprotection, such as preconditioning to cerebral ischemia, causing ischemic tolerance. In the following review we discuss the effects of acute and chronic hypoxia and the release of pro-inflammatory cytokines on synaptic transmission and plasticity in the central nervous system. Specifically we discuss the effects of the pro-inflammatory agent TNF-α during a hypoxic event.

## 1. Introduction

In the central nervous system, hypoxia occurs when there is an inadequate supply of oxygen to neuronal tissue. During acute hypoxia multiple oxygen sensors are deployed allowing neurons to adapt to the response. These responses to hypoxia include synaptic signaling decreases usually as a result of anerobic metabolism changes whilst chronic hypoxia may give rise to more severe perturbations of synaptic transmission and the activation of transcription factors that regulate oxygen homoestasis [1]. Different neurons adapt to a decreased oxygen supply to the brain in many ways, reflecting the diverse role of neuronal functions and also the extent of the hypoxia experienced. It is now known that an hypoxic event in brain tissue can cause ATP to drop by as much as 90% in less than 5 min. Additionally, oxygen-sensitive ion channals including Na^+^ and K^+^ are activated bringing about changes in excitation and inhibition of neuronal and glial cells [2]. Depolarisation of cells may also take place causing the uptake of Na^+^ and Cl^−^ into cells followed by passive influx of water, resulting in swelling and oedema [2]. Hypoxic insults may also activate voltage-gated Ca^2+^ and K^+^ ion channels and glutamate transporters, eventually causing excess glutamate to spill into the synaptic regions causing excitotoxicity. On the other hand, many of the long-term hypoxic responses are mediated by hypoxia inducible factors (HIF), such as HIF-1α [3,4]. HIF-1α is a universally expressed transcriptional mediator of the hypoxic response that is degraded in an oxygen-dependent manner. Under normoxic conditions, HIF-1α has a half-life of approximately 8 min due to hydroxylation by prolyl hydroxyl domains (PHDs) [5]. These PHDs exist in three different isoforms, PHD1, PHD2, and PHD3 and all require oxygen, iron, ascorbate and 2-oxoglutarate, a product of the oxygen dependent Kreb cycle, to hydroxylate HIF-1α. Under hypoxic conditions the Kreb cycle is inhbited leading to a reduction in 2-oxoglutarate, preventing the binding of PHDs to the targeting proline domains [4,6]. During hypoxia, the HIF-1α protein stabilizes allowing it to recruit transcriptional co-activators, which are blocked during normal conditions via factor inhibiting HIF (FIH) [7]. This complex then permits for the transcription of hypoxia-related proteins through binding of the hypoxic responsive element (HRE). HRE binding induces the expression of genes, such as erythropoietin, vascular endothelial growth factor and insulin growth factor. These all play a neuroprotetive role in response to the hypoxic insult.

These acute and chronic responses to hypoxia are clearly manifested during ischemic events in the brain. An example of one such event with a hypoxic component is stroke, which is caused by a reduction in blood flow as a result of an obstruction or rupture of blood vessels within the brain and may cause both acute and chronic episodes of hypoxia. This leads to complex pathological changes taking place, which may lead to tissue necrosis through increased inflammation and oxygen deprivation [8]. During an ischemic stroke the eventual restriction of oxygen in the brain due to an obstruction leads to a cascade of events including hypoxia, increased expression of pro-inflammatory cytokines like tumor necrosis factor alpha (TNF-α) and interleukin-1beta (IL-1β), as well as increased release of the excitatory neurotransmitter glutamate [9]. In this review we will discuss how hypoxia and the release of pro-inflammatory cytokines can effect synaptic transmission and plasticity in the central nervous system (CNS).

## 2. Hypoxia and Synaptic Signaling

Synaptic transmission in the CNS requires approximately 30% to 50% of cerebral oxygen. Therefore many of the changes in the CNS related to acute hypoxia stem from modifications of synaptic excitation and depression. The responses to hypoxia, which occur within seconds, most likely do not involve a role for HIF-1α stabilization. Additionally, upon re-oxygenation after a short period, synaptic transmission can recover to 100% in many brain regions [10]. This decrease in synaptic signaling during acute hypoxia is thought to protect some neurons during ischemic events. Adenosine is one of many neurotransmitters, which plays a vital role in the neuroprotective response to hypoxia [11]. Adenosine A_1_ receptors (A_1_Rs), in particular, play a part in altering neurotransmitter release [12] and have wide expression levels throughout the CNS [13]. This inhibitory neuromodulation by A_1_Rs is coupled to inhibitory G_i_ or G_o_ containing G-proteins [14]. Activation of the receptor stimulates adenylyl cyclase, activates inwardly rectifying K^+^ channels, thus inhibiting Ca^2+^ channels and activation of phospholipase C. This inhibits the release of a number of neurotransmitters including glutamate, dopamine, serotonin and acetylcholine thus making it the primary neuroprotective receptor. Adenosine forms through the enzymatic catabolism of adenosine triphosphate (ATP) into adenosine monophosphate (AMP), which then is broken down by ecto’5 nucelotidases into adenosine (see Figure 1). Adenosine kinase is mainly responsible for the removal of adenosine via phosphorylation to AMP [15]. Under hypoxic conditions when there is a build-up of adenosine in the extracellular space, hypoxia induced factors such as HIF-1α also cause an increase in the ecto’5 nucelotidases CD73, allowing for a breakdown of extracellular ATP into adenosine [16,17].

It is now known that during hypoxia, HIF-1α inhibits the equilibrative nucleoside transporters ent-1/2 located on the membranes of neurons and glia preventing adenosine reuptake into the neuronal cell [18]. Extracellular adenosine binds to A_1_Rs located on both the postsynaptic and presynaptic membranes. Postsynaptic A_1_R activation inhibits the activation of glutamatergic *N*-methyl-d-aspartate receptors (NMDARs) and adenosine binding to A_1_Rs located presynaptically [14]. Inhibition of neurotransmitter release can be suppressed by the addition of an A_1_R selective inhibitor, such as 8-cyclopentyl-1,3-dipropylxanthine (DPCPX), suggesting that adenosine binding is necessary for the reduction of post synaptic potentials [19]. It has also been shown that the A_1_R binding of adenosine inhibits NMDA receptor activation [20]. Creation of knockout mice with the deletion of presynaptic A_1_Rs, uncovered the neuroprotective role that adenosine receptor binding plays in the hypoxic response [21]. Synaptic depression of the excitatory post-synaptic potential (EPSP) was attenuated allowing activation of glutamatergic NMDA receptors and increasing the likelihood for excitotoxicity. More importantly decreased extracellular levels of adenosine have been shown to lead to a loss of hypoxia-induced neuroprotection after repeated exposure to hypoxia [22].

The depression of synaptic transmission in longer term hypoxia goes beyond a neuroprotective role. For example during longer duration hypoxia, nicotinamide adenine dinucleotide phosphate-oxidase oxidase production of reactive oxygen species (ROS) such as superoxides by microglial complement receptor 3 can activate protein phosphatase 2A (PP2A), which causes the internalization of postsynaptic α-amino-3-hydroxy-5-methyl-4-isoxazolepropionic acid receptors (AMPARs) through serine-threonine dephosphorlation [23]. This is similar to the discovery that the oxygen sensing *C-elegans* protein *egl-*9, which regulates HIF in an oxygen-dependent manner can also regulate C. *elegans* glutamate receptor-1 (GLR-1) trafficking through the generation of isoform-specific transgenes which interact with the GLR-1 promoter [24,25]. In normoxic conditions, egl-9 binds to Lin-10 preventing its phosphorylation, this complex then allows for the movement of glutamate receptors to the synapse. Under hypoxic conditions, Lin-10 is phosphorylated, thus preventing the formation of the EGL9/Lin-10 complex leading to a lack of synaptic GluR1 receptors [26].

One particular form of hypoxia, chronic intermittent hypoxia (CIH) may have specific detrimental effects on CNS function. CIH can lead to the over-activation of NMDARs, leading to an overload of intracellular Ca^2+^ and a dephosphorylation of extracellular signal-regulated kinases (ERK) [27]. The CA1 region of the hippocampus is thought to be selectively vulnerable to CIH damage due to the high density of glutamate receptors located on its pyramidal neurons [28]. CIH also leads to a reduction in the levels of the transcription factor cAMP response element-binding protein (CREB) in its phosphorylated form [29]. This reduction in activated CREB leads to a lowering of CREB transcriptional targets, such as brain-derived neurotrophic factor (BDNF), causing cognitive dysfunction [30]. The CIH-induced cognitive dysfunction was shown to be repaired through exogenous application of BDNF to the hypoxic cell [30]. Perinatal hypoxic events may also lead to increases in excitability in hippocampal regions. These events usually occur after asphyxia events just after birth and can lead to long term synaptic changes. Changes in excitability in some local brain regions such as the CA1 region have also been noted [31]. The pursuant neonatal seizures may be related to the phosphorylation of the AMPA GLUA1 receptors on serine 183 and serine 845. This may enhance AMPA receptor excitatory post synaptic currents (EPSCs) which allows for a decrease in the percentage of silent synapses and an increase in AMPA receptor function [32]. This loss of silent synapses is thought to be the mechanism, which attenuates synaptic plasticity in adult life [33]. In critical cases of hypoxia-re-oxygenation the brain loses the ability to form new memories. This anterograde amnesia is decoupled from the hippocampus and its primarily caused by adenosine up-regulation of caspase 1 and then IL-1β in the amygdala [34]. These effects were shown to last up to five hours after re-oxygenation with caspase inhibitors, such as YVAD-CMK, able to shorten the recovery time [34]. The links of hypoxia to cognitive disorders, as well as ability to cause neuronal apoptosis through hyper-excitability, displays the importance of understanding hypoxia and preventing its long-term effects.

## 3. Hypoxia and Synaptic Plasticity

As previously mentioned, hippocampal neuron exposure to hypoxia may lead to cognitive deficits due to synaptic plasticity impairments [35]. Many studies have investigated the relationship between oxygen deprivation and synaptic plasticity. Early studies indicated that brief periods of hypoxia could disrupt long-term potentiation (LTP) in the CA1 hippocampus and that this effect could be reproduced with brief application of adenosine prior to the induction of LTP [36,37,38]. It was later discovered that a brief anoxic episode, as opposed to hypoxia, applied to brain slices, could generate a new type of LTP although still voltage-, NMDA- [39], protein kinase C (PKC)- and NO-dependent [40,41,42]. It is proposed that it is the re-oxygenation and not initial de-oxygenation of neurons and the subsequent high concentration of glutamate that in fact causes the excessive activation of NMDARs and subsequent large influx of Ca^2+^ [43]. It has also been shown that chemically-induced hypoxia with the use of PHD inhibitors, and thus hypoxia mimetics, whilst having no effect on synaptic signaling at low concentrations *per se*, could inhibit LTP in the hippocampus [44,45]. Application of the iron chelator deferoxamine mesylate (DFO) or dimethyloxaloglycine (DMOG), both non-specific pharmacological inhibitors of PHD, and thus increasers of HIF-1α expression [46] could impair LTP in the CA1 hippocampus [4,44,45,47]. Interestingly the application of DMOG to the dentate gyrus region of hippocampal slices did not impair LTP [29]. It is believed that these effects of PHD inhibitors are not HIF-dependent. There is also increasing evidence for a role for CIH in synaptic plasticity and specifically LTP. Initial reports in early 2000 demonstrated that CIH treated animals demonstrated impaired LTP in isolated rat hippocampal slices [48,49]. More recently two reports have put forward evidence for a role for BDNF in this impairment [30,50]. They found that application of BDNF reversed the IH-induced impairment of LTP. In our own laboratories we have implicated a role for PHDs in this inhibition of LTP by intermittent hypoxia [29].

## 4. Hypoxia and Neuroinflammation in the CNS

During an ischemic stroke and resulting hypoxia, inflammatory cytokines are released by microglia, neurons and astrocytes with glutamate largely released by neurons. The up-regulation of pro-inflammatory cytokines through the activation of microglia and astrocytes in the brain contribute a great deal to ischemic brain damage [51]. During hypoxia, HIF-1α binds to HRE like binding sites allowing for the up-regulation of cytokines, such as IL-β, IL-6, IL-8, and TNF-α. Mutations in either the HIF-1α gene or its binding site at the promoter inhibit this cytokine up-regulation [46]. Up-regulation of IL-1β is related to hypoxic hyperexcitability due to the fact that IL-1β can activate tyrosine kinases, which then phosphorylate the NMDAR subunits, NR2A and NR2. This increase in NMDAR potentiation leads to excessive flow of Ca^2+^ leading to hyperexcitability and neuronal injury [52]. Hypoxia also leads to activation of nuclear factor κB (NFκB) signaling pathways whereby HIF-1α has a molecular interaction with the inflammatory mediator NFκB. HRE binding, as seen in Figure 2, allows for the expression of NFκB, which then activates the transcription of inflammatory genes and HIF proteins [53]. NFκB expression is increased when hypoxia is followed by a period of re-oxygenation [54]. Reactive oxygen species (ROS) have been shown to both activate and inactivate NFκB, which could explain the importance of the re-oxygenation period. ROS can trigger both apoptotic and necrotic cell death depending on the severity of the oxidative stress [55,56,57]. Another form of hypoxia, CIH, such as seen in sleep apnea can lead to neuronal cell death and one of the mechanisms involved may be inflammation. Neural inflammation caused by CIH can be region specific with the expression of microglial toll-like receptor-4 (TLR4) increased differentially across areas of the CNS [58]. Hypoxia-re-oxygenation increases microglial levels of inducible nitric oxide synthase (iNOS) leading to neuronal cell loss through apoptosis and memory impairment [59] Many other insults such as bacterial, viral, cytokines and neurodegenerative insults induce iNOS in microglia [60]. This increase in iNOS raises the levels of NO allowing for the inhibition of neuronal respiration causing glutamate release [61]. Rho-associated protein kinase (ROCK) is thought to play a vital role in this pathway as the introduction of the ROCK inhibitor, fasudil, attenuates the neuronal apoptosis [62]. Thus inflammatory pathways and microglial activation are key components to the hypoxic response whereby their activation allows for formation of ROS as well as having the ability to modulate glutamatergic receptors. The important role they play in causing neuronal cell damage as well their potential to be neuroprotective through hypoxic preconditioning makes the inflammatory response a vital therapeutic target in hypoxia. Only recently has it been reported that patients with obstructive sleep apnea (involving episodes of IH) were 1.37 times more likely to have Parkinson’s disease than patients without the disease [63].

## 5. TNF-α and Hypoxia

TNF-α, a pro-inflammatory cytokine produced primarily by monocytes and macrophages in the periphery and microglia and neurons in the CNS, is involved in the promotion of the inflammatory response and cognitive dysfunction [64,65]. TNF-α is initially produced as a 212-amino acid-long type II transmembrane that is stable as a homotrimer. The cleavage of the membrane-integrated form by TNF-α converting enzyme produces a soluble homotrimer, which binds to either of two receptors, TNF receptor type 1 (TNFR1) or TNF receptor type 2 (TNFR2). TNFR1 is constitutively expressed throughout most tissues and is thought to be the main TNF signaling receptor. The activation of TNF-R1 leads to either apoptotic cell death or the activation of either the caspase-8 pathway or c-Jun NH2-terminal kinase (JNK) pathways, or neuroprotection through the binding of IκB kinase (IKK) complex and the subsequent activation of the NFκB pathway [66]. The signaling network in TNF-R1 is interesting due to the extensive crosstalk between the NFκB, and JNK signaling pathways. The cells susceptibility to apoptosis increases in the absence of NFκB. The activation of TNFR2 leads to the activation of the NFκB pathway, phosphatidyl-inositol-3 kinase (PI3K) and subsequent transcription of neuroprotective mediators like calbindin and manganese superoxide dismutase [67,68]. Specifically in microglia activation of TNFR2 anti-inflammatory pathways may be induced [69]. A putative role for TNF-α has been shown in rats infused with lipopolysaccharide (LPS may promote the secretion of pro-inflammatory cytokines including TNF-α and IL-1β) into the fourth ventricle to induce chronic neuroinflammation [70]. TNF-α synthesis inhibition was found to restore the neuronal function as well as reverse cognitive deficits induced by the chronic neuroinflammation [70].

It is becoming apparent that TNF-α is one of the most important inflammatory cytokines to be studied in relation to neuronal damage caused by the absence of oxygen due to the fact that it actively participates in the immune-mediated inflammation of stroke and other neurodegenerative diseases with an hypoxia component [71]. The release of TNF-α is a result of the pathogenesis of disorders such as stroke [72], Alzheimer’s disease [73], Parkinson’s disease [74] and severe infections such as meningitis [75], yet its role during hypoxia is not fully understood. In severe ischemia TNF-α levels appear to be elevated in affected brain tissue after 24 h [76]. One such critical role in neuroinflammation has been illustrated whereby TNF-α can damage dopaminergic neurons and thus anti-TNF agents may ameliorate Parkinson’s disease [74]. Despite many research papers in this field few laboratories have investigated the effects of acute hypoxia and inflammatory mediators on synaptic transmission [77,78]. Recently our laboratory reported that recovery of synaptic transmission in CA1 neurons was impaired post-hypoxia in the presence of TNF-α [77]. It also been shown that HIF-1α has a binding site for the Fas Associated Death Domain promoter, which is an adapter molecule in TNFR1 mediated cell death. Therefore it has a direct role in TNF-α mediated apoptosis which may help explain the poor recovery of EPSPs following a hypoxic insult [79].

A growing body of evidence indicates that TNF-α may play a role in the regulation of tolerance to chronic hypoxia such as occurs in ischemia yet it has a deleterious effect in ischemic brain injury after stroke [80]. It seems that administration of a high dose of lipopolysaccharide (LPS) may induce a robust inflammatory response that can result in lethal septic shock whereas administration of a low dose of LPS may induce a protective state of tolerance to subsequent exposure to LPS at doses that might cause serious injury [81,82]. In fact LPS preconditioning is known to exert neuroprotection from cerebral ischemia [83,84]. In cerebellar granule neurons the neuroprotective effects of LPS preconditioning were said to be independent of endogenous IL-1β but dependent on endogenous TNF-α and also IL-6 [85]. Our laboratories have recently provided evidence that TNF-α has a preconditioning effect following a glutamate toxic insult 24 h later in the CA1 region of rat organotypic slices [65]. We suggested that the preconditioning effects may be as a result of changing resting Ca^2+^ levels and Ca^2+^ influx in the presence of TNF-α.

## 6. TNF-α and Synaptic Plasticity

A growing body of evidence has highlighted the role of TNF-α in glutamatergic synaptic plasticity and scaling. It has been shown that TNF-α has an inhibitory effect on LTP in both the CA1 and dentate gyrus [76,86,87,88,89]. Studies initially carried out by Tancredi *et al.* (1992) [90] showed an inhibitory effect of TNF-α on LTP induction in the CA1 region, which was concentration-dependent. However, they demonstrated that short-term application of TNF-α (>50 min) did not affect LTP. These findings and others highlight the various parameters involved in the regulatory role that this cytokine plays in synaptic plasticity. The inhibitory actions of TNF-α on LTP have been shown to be mediated through the signaling pathways, P38 MAP kinase and JNK [91]. Butler *et al.* (2004) [88] reported that the inhibition of LTP by TNF-α was in fact a biphasic response. SB203580, a P38 MAPK inhibitor, blocked the early inhibition of LTP by TNF-α but did not reverse its late inhibition (3 h following induction), possibly due to the requirement for new protein synthesis. Using antagonists for metabotropic glutamate receptor 5 (mGluR5) and ryanodine, a potential role for metabotropic glutamate receptors and ryanodine sensitive intracellular Ca^2+^ stores in TNF-α mediated inhibition of LTP have also been proposed [87].

Other studies have provided evidence that exogenous application of TNF-α whilst not inhibiting LTP in the CA1 region of the hippocampus may alter homeostatic plasticity (synaptic scaling) rather than synaptic plasticity [92]. These studies have shown that glia released TNF-α is required for synaptic scaling through AMPAR trafficking to the membrane [92,93,94]. Others have reported that the increase in AMPAR expression on the cell surface is mediated through the P13 kinase pathway and the AMPARs trafficked were lacking the GLR-2 subunit. Since LTP is dependent on synaptic glutamate it is also interesting to note that TNF-α has been shown to increase glutamate release from astrocytes [95], block glutamate transporters [96], and also may have a modulatory effect on the expression of GLT-1 and GLT-2. These effects combined may result in increased glutamate concentrations in the synaptic cleft [97,98]. TNFR1, but not TNFR2, may play an important role in AMPAR localization on the membrane of cortical neurons. Deletion of TNFR1 resulted in a decrease of AMPAR clustering on the synaptic membrane, which was not rescued by exogenous application of TNF-α [99]. These observations indicate a potential therapeutic approach for TNF-α via TNFR1 in mediating AMPAR excitotoxicity. Glutamatergic gliotransmission is an important stimulatory input to excitatory synapses and it has been shown that TNF-α is a modulator of this process in the dentate gyrus [100]. Many of the discrepancies observed with regard to the effects of TNF-α on LTP may be region specific or indeed depend on the induction protocol used to induce LTP. There are many factors regulating the magnitude of LTP induced by different parameters such as high frequency stimulation and theta burst stimulation [101] (Figure 3). Recently, we have shown that the stimulation parameters used to induce LTP may have an influence on TNF-α’s ability to inhibit LTP [102]. TNF-α has no inhibitory effect on LTP when induced with prolonged high frequency stimulation (HFS) whereas full inhibition was observed when LTP was induced by theta burst stimulation (TBS). Figure 3 illustrates a potential mechanism that might explain this discrepancy whereby TBS may trigger alternative signaling cascades to HFS that can be modulated by TNF-α.

## 7. TNF-α, Hypoxia and Synaptic Plasticity

Hippocampal slices exposed to acute hypoxia may recover when oxygen is re-introduced. Recently it has been shown that in the presence of TNF-α there is an impairment in the recovery of synaptic transmission in the CA1 region post-hypoxia [77]. Conversely, hypoxia has also been shown to increase intercellular Ca^2+^ levels and activate calmodulin-dependent protein kinase II (CaMKII) through a TNF-α independent mechanism [103]. However CaMKII is also capable of activating the PI3K-PKCλ-AMPAR signaling pathway. TNF-α has been found to play roles in cell adhesion up-regulation, disruption of the blood brain barrier and is a key component for the participation of glial cells in the physiological control of synaptic transmission and plasticity through the release of glutamate, a process known as glutamatergic gliotranmission [100,104]. TNF-α has been shown to increase the release of glutamate from astrocytes, maintain glutamate levels through the blocking of glutamate transporters [96] and modulate the expression of Glut-1 and Glut-2. All these effects by TNF-α result in the increase in the concentration of glutamate in the synaptic cleft, which may have an influence on the magnitude of LTP post-hypoxia. Using a robust LTP-inducing stimulus protocol we have been able to demonstrate a significant enhancing effect of TNF-α on LTP post hypoxia but only in the dentate gyrus of the hippocampus [102]. In the presence of DMOG (a non-specific PHD inhibitor) this enhancement of LTP was still evident perhaps indicating a novel HIF/PHD-independent effect of TNF-α [102].

## 8. Conclusions

Hypoxia is one of the key components, which can arise from neuropathological conditions such as stroke, Parkinson’s or Alzheimer’s disease. Hypoxic events can cause the release of pro-inflammatory cytokines from neurons and glial cells, such as TNF-α, which can lead to further neurotoxicity or indeed neuroprotection in the brain. However, the effects of TNF-α on neurons during de- and re-oxygenation of neurons is largely unknown. Many studies have now shown that pro-inflammatory cytokines, such as TNF-α, play a key role in the regulation of synaptic transmission and plasticity in the absence and presence of acute hypoxia, especially within the hippocampus. The mechanisms by which elevated levels of TNF-α have an enhancing or detrimental effect on synaptic signaling and synaptic plasticity in the presence or after a hypoxic event remains to be elucidated.

## Figures and Tables

**Figure 1 brainsci-06-00006-f001:**
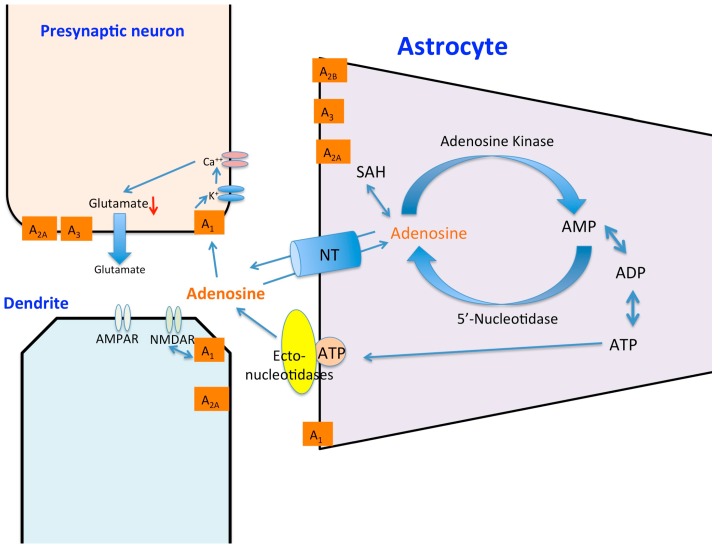
The effects of hypoxia on adenosine release in the CNS. Hypoxia causes a breakdown of extracellular ATP and AMP along with activation of membrane-bound transporters such as ectonucleotidases, leading to a build-up of extracellular adenosine. Adenosine binds presynaptically to A_1_Rs attenuating voltage dependent calcium channel (VDCC) function and thus neurotransmitter release and also binds postsynaptically to A_1_Rs receptors inactivating glutamatergic NMDARs. Adenosine is released from astrocytes in response to chronic hypoxia.

**Figure 2 brainsci-06-00006-f002:**
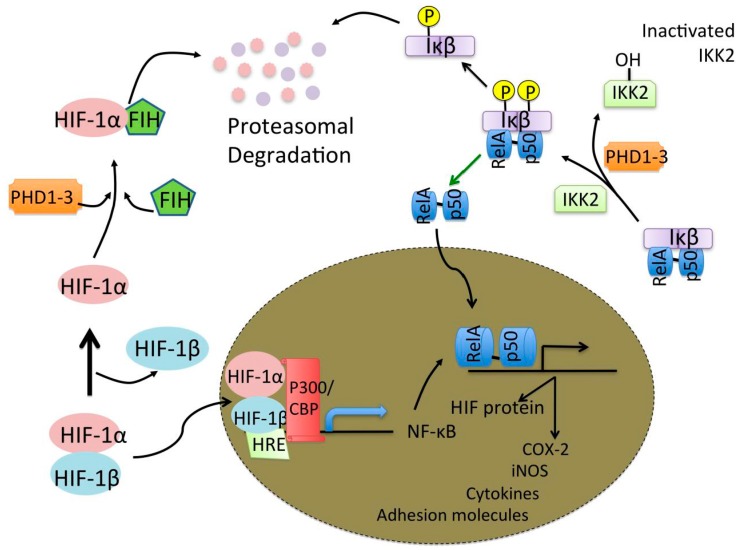
Hypoxia and NFκB activation. During hypoxic HIF-1α binding to the HRE induces the expression of NFκB (**left**). NFκB p50 p65 dimer is able to freely activate the transcription of inflammatory and HIF proteins (**right**).

**Figure 3 brainsci-06-00006-f003:**
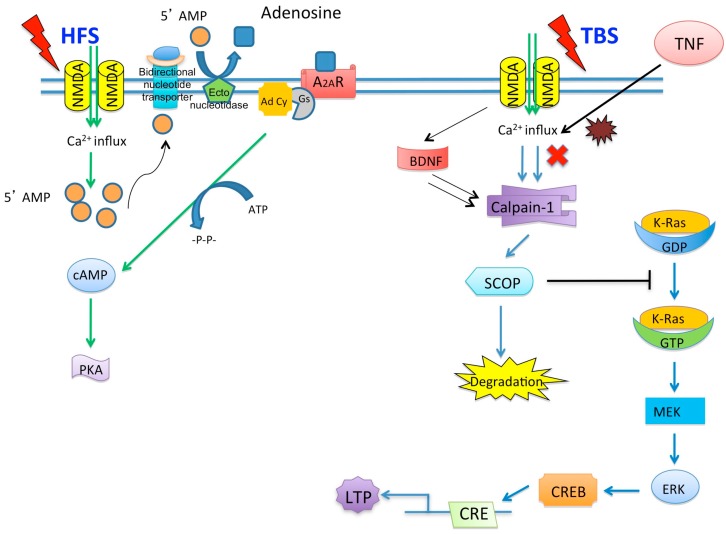
Putative signaling pathways activated after HFS- and TBS-induced LTP. HFS-induced LTP may be dependent on the breakdown of 5′ AMP into adenosine. Adenosine activates the A_2A_R receptor leading to cAMP and PKA activation. TBS-induced LTP involves the influx of Ca^2+^ and subsequent activation of calpain-1. The activation of calpain-1 leads to a calapin-1-mediated suprachiasmatic nucleus circadian oscillatory protein degradation and ERK activation. Exogenous TNF-α inhibits LTP induced by TBS only. During hypoxia, TNF-α may have potentiating effect on HFS-induced LTP but not TBS.

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
