# Peer review of "The Effects of Hypoxia and Inflammation on Synaptic Signaling in the CNS"

_brainsci, 2016, doi:10.3390/brainsci6010006_

Round 1

Reviewer 1 Report

In the manuscript “The effects of hypoxia and inflammation on synaptic signaling in the CNS” The authors have presented an detailed and scientifically sound review about the current knowledge available on this topic.

Minor issues:

There are many abbreviations used in the text, a table/list summarising the abbreviations will be helpful for the reader.

At many places in the text symbol alpha is missing (after TNF and HIF-1)

Author Response

Reviewer 2

In light of referee1’s comments we have re written large sections of the review

Minor issues:

There are many abbreviations used in the text, a table/list summarising the abbreviations will be helpful for the reader.

An abbreviation section has now been inserted although some sections have been deleted that had many of these original abbreviations

At many places in the text symbol alpha is missing (after TNF and HIF-1)

We have corrected all of these.

Reviewer 2 Report

The manuscript entitled “The effects of hypoxia and inflammation on synaptic signaling in the CNS” by Mukandala et al. discusses the role of HIF, inflammatory cytokines and adenosine in signaling pathways involved with hypoxia and inflammation. Despite the excellent laboratory from which this paper comes, it is poorly written and requires significant reorganization and focus. In general, the paper does not flow well, and it is not clear what point the authors are trying to make overall. The sections are very choppy, and one section doesn’t necessarily follow logically to the other. The abstract is not very informative. Much of the text reads like a laundry list of facts with little synthesis and extrapolation to how the ideas fit together, or support whatever the point is supposed to be. There is way too much detail on HIF and TNFR signaling, and the extrapolations of certain hypoxia paradigms to sleep apnea are incorrect. A few of the most significant issues are elaborated below, but there are others.

The very first thing that is discussed in the text is the detailed intricacies of HIF regulation, but it is not clear why this information is provided with such detail when most of the molecules referenced there are never discussed again. And why the focus is on HIF, as opposed to the many other transcription factors that also contribute to the hypoxic response is also not clear. For readers with specific interests in HIF, they can consult them any available HIF reviews. This article is supposed to be about hypoxia and inflammation- it would be better if the very first sentence of the abstract isn’t ischemia/stroke since the article’s focus is hypoxia and inflammation.

The second section is about adenosine- again- it is not clear why adenosine, and not the many tens of other neurotransmitters that could have been discussed. And this comes right after a whole section on HIF and gene transcription. There is no transition or logical rationale that gets us to adenosine. In this regard, if there is mention that adenosine production by astrocytes in response to hypoxia is controversial, then one must present both sides of the controversy, not just one.

There are several statements that require references, particularly in the section on hypoxia and neuroinflammation. ROS activating NF-kB requires a reference, and it should also be mentioned here that ROS have also been shown to decrease NF-kB binding just as well. Neither NFkB or iNOS is on the cell surface. The last sentence on p4 also needs a reference, but is worded more matter of factly than is accurate- other CNS cells do express iNOS- for example, in CIH, iNOS is upregulated in neurons... so this statement is not necessarily accurate. It is also confusing what is meant by activating “NF-kB on the microglial cell surface”… how can this happen? This statement also requires a reference.

At the top of p6 where references 44 and 45 are discussed. It makes it seem like sleep apnea patients should be epileptic/seizure prone, which is obviously not the case. The in vivo paper used 15 min durations of 4% O2 for 15 mins each, twice- this is not even close to sleep apnea or sleep disordered breathing. OSA patients would never be apneic for 15 mins (without dying), and their hemoglobin would never desaturate to the levels those mice were experiencing. It would be safer, and more correct, not to try to relate all models of hypoxia delivered episodically, to sleep apnea.

In the TNFa section, why does the fact that TNFR2 is strongly regulated mean that it plays a role in the lymphoid system? In the last line of the last paragraph in this section, the examples provided (LPS, AD) have nothing to do with hypoxia, at least that was provided. The relationship between those stimuli and hypoxia should be elaborated, or CNS disease examples that involve hypoxia should be used to be more supportive of this sentence saying “…neuronal damage caused by absence of oxygen.” Also, description of hypoxia in PD (on p10) should also be included.

In the TNF in hypoxia section- it is not clear how we transitioned from TNF in stroke, AD and meningitis to acute hypoxia in synaptic transmission. Up until now the hypoxia that has been discussed was detrimental- an aspect of ischemia/stroke. But with those references to LTF, it is being used as a stimulus to promote neuroplasticity- however, this nuance is not made clear, nor is how TNF fits in with that form of neuroplasticity.

On the bottom of p8, when discussing reference 93, that whole argument is very confusing and should be clarified. If LPS pre-conditioning is thought to be beneficial because it upregulates TNF production, how does LPS decrease TNF in ischemia? The sentence says “…and that the reduction in TNFa signaling following ischemia conferred neuroprotection.”

Author Response

Reviewer 1

The manuscript entitled….

Thank you for all of your helpful comments on the review. We have now had a chance to go through your comments in detail and agree that the manuscript required some re-editing and re-writing. We have deleted many sections and added some text. We also deleted 2 of the 5 figures.

We agree that the abstract needed attention and have re-written it accordingly. We now put in a little more detail on the contents of the review and the points that we are addressing. The abstract finishes with an outline of the review.

We have now taken out most of the detail about HIF and adenosine (see below). We have also changed the order in which adenosine appears in the review. We have also re-written much of the sleep apnea sections and deleted other parts.

Detailed intricacies of HIF regulation…..

The referee is indeed correct. We did not need to concentrate on HIF to the extent we did, as there are many reviews in the HIF field and it is not the primary focus of this review. We have now shortened this section and refer the reader to reviews in the area. This section is now integrated into the ‘Introduction’.

We also refer to hypoxia at the beginning of the abstract rather than stroke or ischemia. Finally we have left out the figure on HIF regulation (old Figure 1) as this is not now appropriate.

Why adenosine and not the many tens of other neurotransmitters….

We wrote about adenosine as we have some data on its effect on synaptic transmission but entirely agree with the referee that there are many other neurotransmitters that could be discussed. We therefore have reduced this section significantly and overall integrated the section into ‘The role of hypoxia in synaptic signalling’. We also refer to some of the other neurotransmitters involved in hypoxia. We have deleted the statements about astrocytes hypoxia and controversy.

There are several statements that require references……

We apologise for the poor section on ROS and NFkB. We have rewritten this accordingly and agree of course that neither NFkB nor iNOS are on cell surfaces. This sentence was drawn up incorrectly/sloppy and we were of course not implying that only microglia express iNOS. We have also deleted some of this section. We mention that ROS can both increase and decrease NFkB binding and include appropriate references. We have excluded some current references and included some new references.

References 44 and 45, Sleep apnea p6…..

We agree with the referee that these two sentences are confusing and may not be a good correlate to sleep apnea in humans. We have therefore deleted these two sentences from the review and appropriate references. The rest of the section has been re-written.

TNF section and TNFR2…..

Reference to ‘lymphoid’ is misleading and we have deleted this sentence. We also agree that reference 77 referring to Alzheimer’s disease may not be relevant and have deleted the sentence.  We have re-written the sentence beginning ‘LPS infusion…….’ We have also inserted a few sentences on Parkinson’s disease and a reference.

TNF hypoxia section…..

We agree this section was too technical and long. We have now reduced this section leaving out much of the background on TNF receptors and abbreviations.  We also don’t believe reference to the LTF work adds to the review and have deleted reference to it.

Bottom of P8, reference 93

These sentences were indeed confusing and incorrect. We have now attempted to clarify their content. It seems that administration of a high dose of LPS may induce a robust inflammatory response that can result in lethal septic shock whereas administration of a low dose of LPS may induce a protective state of tolerance to subsequent exposure to LPS at doses that might cause serious injury (Fan and Cook, 2004). [89]. This has been inserted and the previous sentence deleted.

Finally in making all of these changes we have decided to delete figures 1 and 4 so focusing the manuscript more on hypoxia and inflammation in the CNS. We apologise for the poor structure of the previous manuscript and thank the referee for the opportunity to make these changes.

Round 2

Reviewer 2 Report

The mansucript flows much better, and the overall point is now clear. The accuracy of the information is now also much improved.